# Small-rotative fixed-target serial synchrotron crystallography (SR-FT-SSX) for molecular crystals
Sam G. Lewis [1,2], Ben A. Coulson[1], Anna J. Warren [2], Mark R. Warren [2] ✉ & Lauren E. Hatcher [1] ✉

The increasing availability of ultrabright Light Sources is facilitating the study of smaller crystals at faster timescales but with an increased risk of severe X-ray damage, leading to developments in multi-crystal methods such as serial crystallography (SX). SX studies on crystals with small unit cells are challenging as very few reflections are recorded in a single data image, making it difficult to determine the orientation matrix for each crystal and thus preventing the combination of the data from all crystals for structure solution. We herein present a Small-Rotative Fixed-Target Serial Synchrotron Crystallography (SR-FT-SSX) methodology, in which rotation of the serial target through a small diffraction angle ($\varphi$) at each crystal delivers high-quality data, facilitating ab initio unit cell determination and atomic-scale structure solution. The method is benchmarked using microcrystals of the small-molecule photoswitch sodium nitroprusside dihydrate, obtaining complete data to $d_{min} = 0.6$ Å by combining just 66 partial datasets selected against rigorous quality criteria.

With the advent of "fourth-generation" synchrotron sources[1], the numerous projects to upgrade "third-generation" facilities[2–4] and as further X-ray Free Electron Laser (XFEL) facilities come online, the widespread availability of ultrabright X-rays is generating new research including time-resolved studies at femtosecond to attosecond timescales[5–8] and coherent diffractive imaging of individual particles[9,10]. Such developments enable the exciting possibility of studying the structural dynamics of fundamental electron transfer processes in real-time, as increases in X-ray brilliance can facilitate the collection of diffraction data suitable for atomic-scale structure determination even from very small crystals. However, such X-ray brilliance is accompanied by the increased likelihood of severe radiation damage and typically only a portion of the data required for 3D structure determination can be collected on an individual crystal. To address this challenge, multi-crystal methods are developing that aim to "out-run" damage processes by supplying a stream of crystals into the X-ray beam, with only a single diffraction image recorded on each crystal. By recombining the "single shot" images from 100s-1000s of different crystals, in random orientations, these serial crystallography (SX) methods can provide a complete 3D structure of the target material before the effects of radiation damage become significant.

To-date, SX has been pioneered by the macromolecular crystallography (MX) community with progress initially driven by the advent of XFELs[5,11,12], although serial synchrotron crystallography (SSX) studies are increasingly reported[13–20]. SX on organic and organometallic "small-molecule" crystals and hard inorganic samples is less established, although a handful of recent XFEL studies show that interest in SX amongst chemical crystallographers is growing[21,22]. A key challenge for the study of molecular crystals is the relatively sparse diffraction pattern they produce, as their crystal structures typically have much smaller unit cell dimensions compared to proteins. This means that there are typically not enough reflections in the single diffraction image recorded on each crystal to allow indexation of its diffraction pattern, which prevents determination of the orientation matrix for individual crystals and makes it challenging to combine the data from multiple crystals into a cohesive dataset for structure solution. Recent studies have attempted to address this issue, for example by using a shorter, higher energy X-ray wavelength to compress the diffraction pattern and maximize the number of diffraction spots recorded per image, and by studying chemical samples that display relatively large unit cells e.g. metal-organic frameworks[22], and/or by studying systems whose unit cell parameters are already known[23]. Alternatively, using a polychromatic white or pink X-ray beam for a Laue diffraction approach typically generates a larger number of reflections from which to index from, but with the associated challenges of Laue data processing[24]. A recent study by Schriber et al. utilizes a powder diffraction approach, aggregating images with sparse diffraction to produce powder X-ray diffractograms and then indexing these patterns by a graph-theory approach[21]. Whilst such solutions may be effective for particular samples, there remains a need for other methods that facilitate ab initio 3D structure determination of a more comprehensive range of organic and inorganic materials by SX.

[1]School of Chemistry, Cardiff University, Main Building, Park Place, Cardiff, CF10 3AT, UK. [2]Diamond Light Source, Harwell Science and Innovation Campus, Fermi Avenue, Didcot, Oxfordshire, OX11 0DE, UK. ✉e-mail: mark.warren@diamond.ac.uk; HatcherL1@cardiff.ac.uk

We herein report a Small-Rotative Fixed-Target Serial Synchrotron Crystallography (SR-FT-SSX) approach to study samples with small unit cells using monochromatic synchrotron radiation, which has been successfully used to solve the structure of the inorganic photoswitch sodium nitroprusside dihydrate (SNP.2H$_2$O) ab initio, without the need for any prior knowledge of the crystal structure. To the best of the authors' knowledge, this is a novel approach for the study of small molecule crystals, although an analogous serial oscillation methodology for protein samples is previously reported[25]. Our approach uses a fixed-target SX methodology that incorporates a small (5°) rotation about the $\varphi$-axis, such that a series of 0.1°-wide diffraction images are collected at each crystal position. This thin-slicing $\varphi$-scan approach has the advantage of providing adequately sampled reflection profiles, rather than the partial reflections obtained from the static images typically collected by traditional SX approaches. This provides more accurate peak positions and profiles for indexing and integration, facilitating improved data processing and merging of the data from all crystals. In this article, we show how the partial datasets collected on each crystal can be indexed ab initio to determine unit cell parameters and an orientation matrix for diffracting crystals in the grid. The method generates X-ray structures of SNP.2H$_2$O with excellent signal-to-noise from the combination of as few as 28 partial datasets, with fully complete data achieved by combining 66 datasets. A semi-automated data processing pipeline is also outlined, which provides a rapid and visual quality assessment of all sample positions in the fixed-target grid for facile selection of the best datasets to combine into the final structure. Finally, the unit cell parameters determined ab initio in the first round of processing can be further refined by a second iterative processing step, leading to an increase in the quality of the final structure model. This iterative procedure provides a final dataset that is of comparable quality to that collected on an individual single-crystal, proving the viability of unknown small molecule crystal structure determination by SR-FT-SSX.

## Results and discussion
### Sample delivery (fixed-target approach)
The majority of SX studies at XFELs utilize a liquid jet method for sample delivery, by which a slurry of microcrystals suspended in a carrier liquid is injected into the X-ray beam[26]. However, liquid jet approaches can encounter problems including equipment-based issues, e.g. difficulties in flow rate regulation, clogging/blockages in the injector nozzle, and background contribution from the carrier fluid; or sample-based issues, e.g. crystal attrition on injection, poor chemical compatibility between the sample and carrier fluid, and high crystal consumption rates that preclude the study of precious samples. The latter two factors are particularly challenging for chemical targets, which necessitates a different approach[22]. Fixed-target methods have developed as an alternative[27], particularly for synchrotron serial crystallography (SSX) experiments[14,15,17,28]. Crystals are loaded into a multi-well grid mount such that there is, optimally, an individual crystal per sample well, before the grid is mounted at the sample position and an XY rastering procedure is used to expose each well position sequentially to the X-ray beam[14]. In this work we have utilized silicon nitride fixed-target grid mounts to deliver crystals into the synchrotron X-ray beam on Beamline I19 at Diamond Light Source (Fig. 1a). The mounts contain a single "city block" containing a 20 × 20 grid of 400 individual sample wells[28]. As well as minimizing the total sample volume required to only a few microlitres per grid loading, fixed-target grid mounts have the additional advantage that the grids can be assessed visually via microscopy techniques after loading, providing a greater understanding of the distribution of crystals across the grid, and their orientation within the wells, in advance of the SSX experiment. The 400-well grids are comparatively small but have the benefit of increasing the efficiency of the SR-FT-SSX method, in which more data is obtained from fewer crystals as a result of the $\varphi$-scan approach. For future applications, these smaller grids (with external dimensions of 9.2 × 4.2 × 0.3 mm) offer compatibility with liquid nitrogen flow devices for temperature control and with robotic sample handlers for remote handling.

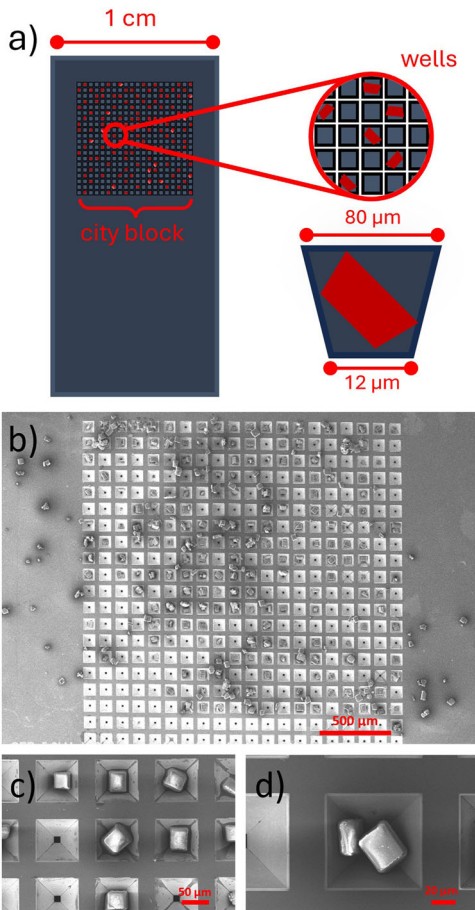

**Fig. 1 | Crystal loading into the fixed target serial grid. a** Schematic description of the components in a fixed-target grid, and images of a fixed-target grid taken by scanning electron microscopy (SEM) to show: **b** the entire grid (×35 zoom), **c** a section of wells containing a single crystal (×270 zoom), and **d** a single well occupied by two crystals (×600 zoom). The grids were loaded using a 2.5 μL aliquot of a [0.08 g/g] suspension of crystals in 1-butanol.

### Sample preparation (grid loading)
In step 1 of the SR-FT-SSX workflow (Scheme 1), microcrystal batches of SNP.2H$_2$O were produced by a batch antisolvent crystallization method. For each batch, a small portion of crystals was analyzed by FTIR spectroscopy and powder X-ray diffraction, confirming the sample to be the known dihydrate form of SNP (Supplementary Note 1)[29]. Direct use of the crystals suspended in their mother liquor (10:1 acetonitrile / deionized water) for grid loading led to rapid, uncontrolled crystallization in-situ on the grid. To address this, a stock crystal suspension was produced in which crystals were suspended in a suitable antisolvent. The selection of an appropriate antisolvent was found to be crucial for evenly dispersed grid loadings and is chosen such that (i) it is a true antisolvent, meaning the crystals will not dissolve even partially, (ii) the material does not react with the antisolvent, enabling long-term storage of the suspension, and (iii) the antisolvent is of a suitable viscosity and density that the crystals become evenly suspended on agitation. This latter point is critical to enable even distribution of crystals across the grid. For SNP.2H$_2$O, of a selection of antisolvents tested, 1-butanol was the most suitable (Fig. S3) and the suspensions produced were stable for a period of 6 months (Fig. S4). The crystallization and 1-butanol suspension procedures are detailed in the Methods and Supplementary Note 2.

Suspensions containing different concentrations of crystals were first tested to ascertain the ideal crystal:antisolvent ratio for grid loading. Of those tested, a 0.08 g/g crystal/antisolvent suspension was found to be most suitable for SNP.2H$_2$O. For the grid loading procedure, an even suspension was

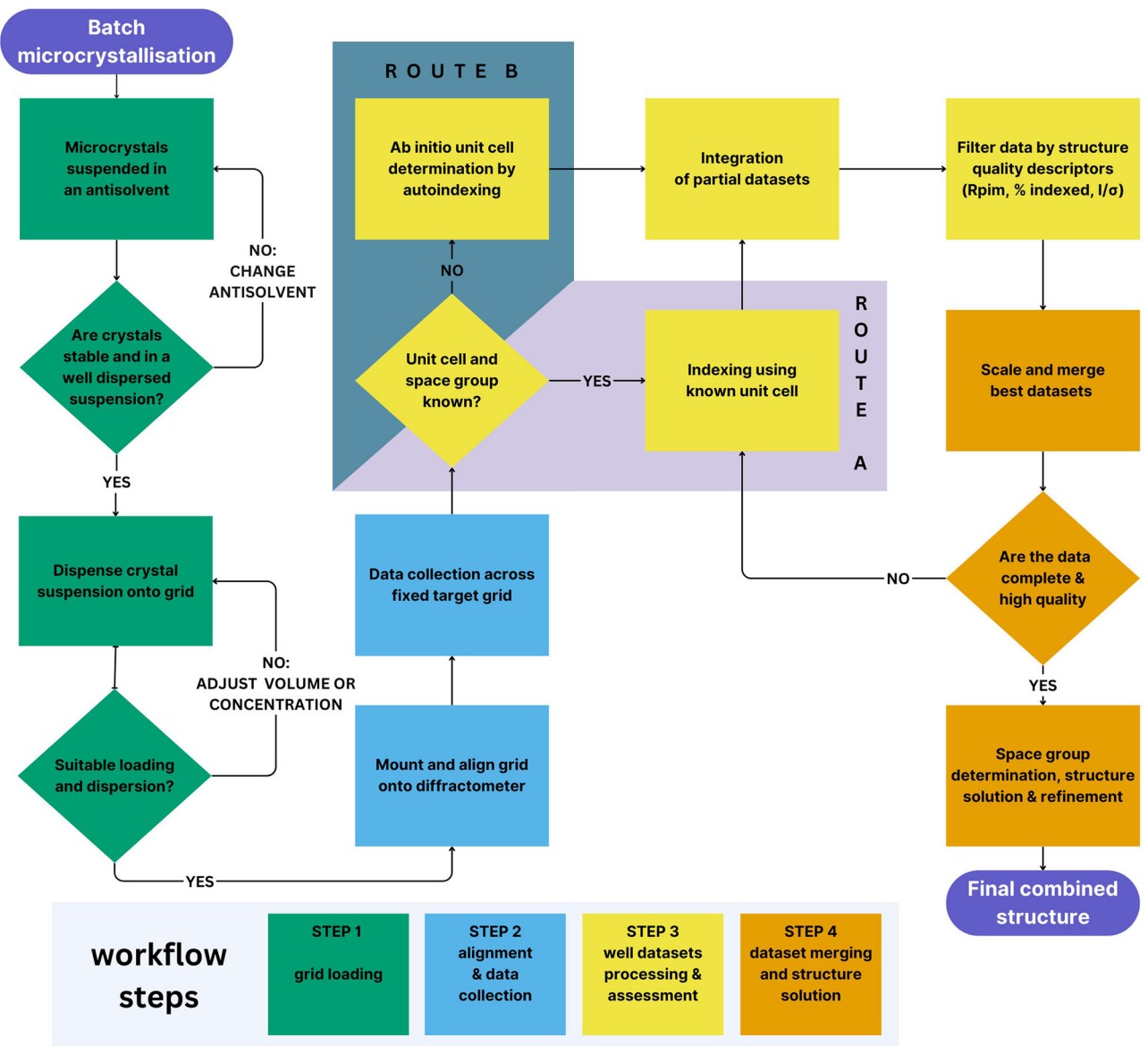

**Scheme 1 | SR-FT-SSX Workflow**. Flow chart describing the SR-FT-SSX method workflow for the study of known and unknown crystal systems.

generated by agitating the microcrystals by repeated inversions of the sample tube before 2.5 μL of the suspension was rapidly withdrawn and pipetted onto the center of the 400 well city block. The pipette tip was used to carefully spread the suspension evenly across the grid surface (Fig. S5) and the crystals allowed to settle into the grid wells. Any excess 1-butanol antisolvent was removed by gentle vacuum filtration using a bespoke 3D printed grid holder (Fig. S6).

Loaded grids were analyzed by polarized light microscopy and scanning electron microscopy (SEM) to visualize the sample loading. The crystallization experiment was tailored to target microcrystals on the order ~40 μm³ so that they will fit comfortably in the wells, which have an inverted pyramidal geometry with a base width of 12 μm and an opening of 80 μm (Fig. 1a). This rational crystal-to-grid size-matching is clear from SEM imaging, which shows the dispersion of crystals across the grid (Fig. 1b) and that a large proportion of the wells are singly occupied (Fig. 1c). As expected, there remains some variability across the grid and some wells are either empty or occupied by multiple crystals (Fig. 2d). Whilst the data processing protocols described here-in are capable of analysing diffraction patterns from multiple crystals in the same well, it is preferable to aim for a lower crystal loading across the grid to promote fewer instances of multiple well occupation.

## Data collection

For data collection (step 3 in Scheme 1), the fixed-target grid was mounted at the diffractometer sample position using a modified magnetic base (Fig. 2a, b). To provide sufficient diffraction data for the indexation of unit cell parameters from each suitable SNP.2H₂O crystal, a unique small rotative approach was implemented during the data collection. At each of the 400 well positions, the serial target was rotated through 5° in $\varphi$, whilst a series of 50 diffraction images of $\Delta\varphi = 0.1°$ and exposure time 0.1 s were collected. A movie showing the small rotation is provided in the Supplementary Information (Supplementary Movie 1). XY rastering of the 400 well city block across the synchrotron X-ray beam was achieved using piezo motor stages to implement a traditional "snake scan" across the serial chip, starting at the top left corner well position (as shown in Fig. 2c). For efficiency of the diffractometer motions and to minimize overheads, the direction of the $\varphi$-scan is reversed at each well position. Full details of the data collection parameters are provided in the Methods and Supplementary Note 3.

This data collection procedure provides a partial dataset of 50 images from each crystal, containing a sufficient number of reflections for indexing. As mentioned above, the reflection profiles are well-sampled by this thin-slicing approach, providing accurate diffraction peak positions. This methodology is more similar to a standard small molecule X-ray diffraction

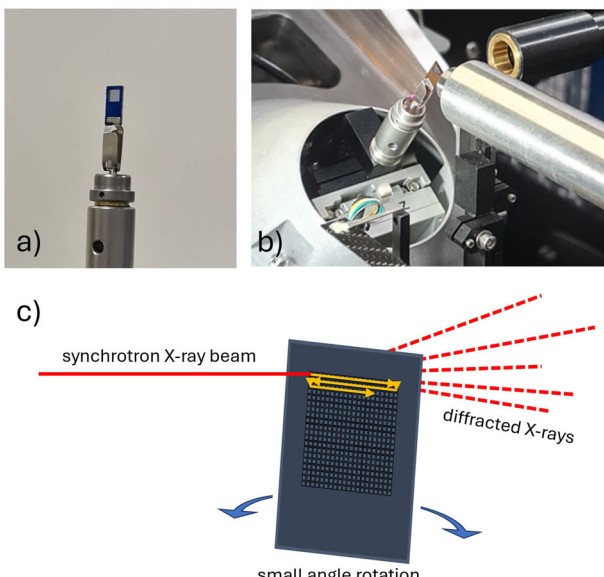

**Fig. 2 | Fixed-target grid mounting on the synchrotron beamline.** Images of the fixed-target serial grid mounted on a modified magnetic base and on the diffractometer in Experimental Hutch 1 on Beamline I19 at Diamond Light Source. **a** Serial chip viewed from below the sample position, **b** serial chip view from side-on, **c** illustration of the XY rastering and small angle rotation procedure used during the FT-SR-SSX data collection.

experiment on a single-crystal and thus delivers unit cell parameters with low estimated standard deviations (Table 1). The total X-ray exposure time on each crystal is 5 s, which was suitably short in this study as the SNP.2H$_2$O crystals are robust. For other experiments a 5 s exposure could be too long, e.g. for samples prone to radiation damage, however, the benefit of the SR-FT-SSX method is that the rotation width can easily be varied (with rotations of $0° < \varphi < 50°$ viable) and, where diffraction quality noticeably decreases across a collection, data-images recorded at longer exposure times can be omitted as required during post-processing.

## Data processing
The full data collection procedure generates 20,000 images that, due to the variability of crystal loading across the chip (Fig. 1c, d), may contain either diffraction from a single crystal, from multiple crystals, or no indexed reflections at all due to an empty well position. The goal for efficient data processing is to quickly sort through the images to identify those containing usable diffraction data, then to merge and scale only the best partial datasets to produce the highest quality crystal structure.

Data processing (Step 4 in Scheme 1) was performed using an autoprocessing pipeline coded in Python 3.1 and could be initiated concurrently with the data collection to provide rapid, on-the-fly assessment of the diffraction data. The partial datasets containing 50 images collected at each sample position are initially processed as separate collections. Spot-finding and indexing procedures are initiated on each partial dataset, using the program DIALS implemented via Xia2[30], and possible unit cell parameters are output for every successful well. Indexing outputs can be used as an initial quality indicator, as any well containing either no crystal or multiple crystals is likely to fail at the indexing stage. Once suitable unit cell parameters and a space group are determined, integration and scaling procedures are then initiated for all viable partial datasets within DIALS[30], providing intensity data for individual wells. Once partial dataset processing procedures are complete, the processing pipeline generates a summary file containing relevant structure quality factors for all partial datasets including indexed unit cell parameters, % of spots matching the unit cell, assigned space group, diffraction signal-to-noise ($I/\sigma(I)$), and residual ($R$) factors. A set of visual data plots of these quality descriptors is also generated (Fig. 3) for quick visualization of the most successful partial datasets. In particular, spatially-resolved plots showing the

## Table 1 | SR-FT-SSX crystal data

| Partial dataset information | | |
|---|---|---|
| | 1 | 2 |
| No. of combined datasets | 28 | 66 |
| $R_{pim}$ cut-off | 5.0 | 5.0 |
| % indexed cut-off | 50.0 | 50.0 |
| $I/\sigma(I)$ cut-off | 6.0 | 6.0 |
| Resolution range (Å) | 9.45–0.60 | 9.57–0.60 |
| **Final merged dataset information** | | |
| Empirical formula | Na$_2$[FeC$_5$N$_6$O]·2H$_2$O | Na$_2$[FeC$_5$N$_6$O]·2H$_2$O |
| Formula weight | 297.95 | 297.95 |
| Wavelength (Å) | 0.4859 | 0.4859 |
| Temperature / K | 298 | 298 |
| Crystal system | orthorhombic | orthorhombic |
| Space group | $Pnnm$ | $Pnnm$ |
| $a$ (Å) | 6.1998 (1) | 6.2005 (1) |
| $b$ (Å) | 11.8966 (1) | 11.8972 (1) |
| $c$ (Å) | 15.5604 (2) | 15.5595 (1) |
| Volume (Å$^3$) | 1147.68 (3) | 1147.802 (7) |
| $Z$ | 4 | 4 |
| Total reflections | 11421 | 27049 |
| Unique reflections | 10742 | 25530 |
| Mean $I/\sigma(I)$ | 26.2 | 35.0 |
| $R_{int}$ (%) | 4.50 | 4.45 |
| $R_{merge}$ ($I$) | 0.045 | 0.045 |
| $R_{meas}$ ($I$) | 0.051 | 0.047 |
| $R_{pim}$ ($I$) | 0.022 | 0.014 |
| $CC_{1/2}$ | 0.998 | 0.999 |
| Redundancy | 4.0 | 8.5 |
| $R_1$ (%) [$I > 2\sigma(I)$] | 2.84 | 2.38 |
| $wR_2$ (%) [all data] | 8.42 | 7.16 |
| GooF on $F^2$ | 1.114 | 1.103 |
| Completeness (%) | 95.3 | 99.3 |
| Largest peak/hole (eÅ$^{-3}$) | 0.39/0.340 | 0.39/−0.29 |

Crystal data for Structures **1** and **2**, collected by the SR-FT-SSX experiment and processed via different routes through the autoprocessing pipeline.

statistics overlaid with their serial grid position in real space are highly-informative (Fig. 3a–c) and can be compared to microscopy analysis to identify the microcrystals that produce the best quality diffraction. Finally, the output statistics file can be used to select the most successful partial datasets to be merged for crystal structure solution. Quality factors of $R_{pim}$, % of spots indexed and $I/\sigma(I)$ were chosen as the most suitable to be used as selection criteria for SNP.2H$_2$O, however other quality-factors are additionally output and may be considered for different materials (see Supplementary Note 5). Once identified, the best partial datasets are re-submitted to DIALS to facilitate multi-dataset scaling and merging procedures through DIALS.scale[30], to produce a final set of combined diffraction intensity data in the required format for structure solution and refinement.

As outlined in the SR-FT-SSX workflow (Scheme 1), two possible processing pathways can be utilized depending on the amount of prior information known about the material. For known crystal structures Route A can be followed, in which the known unit cell parameters can be provided

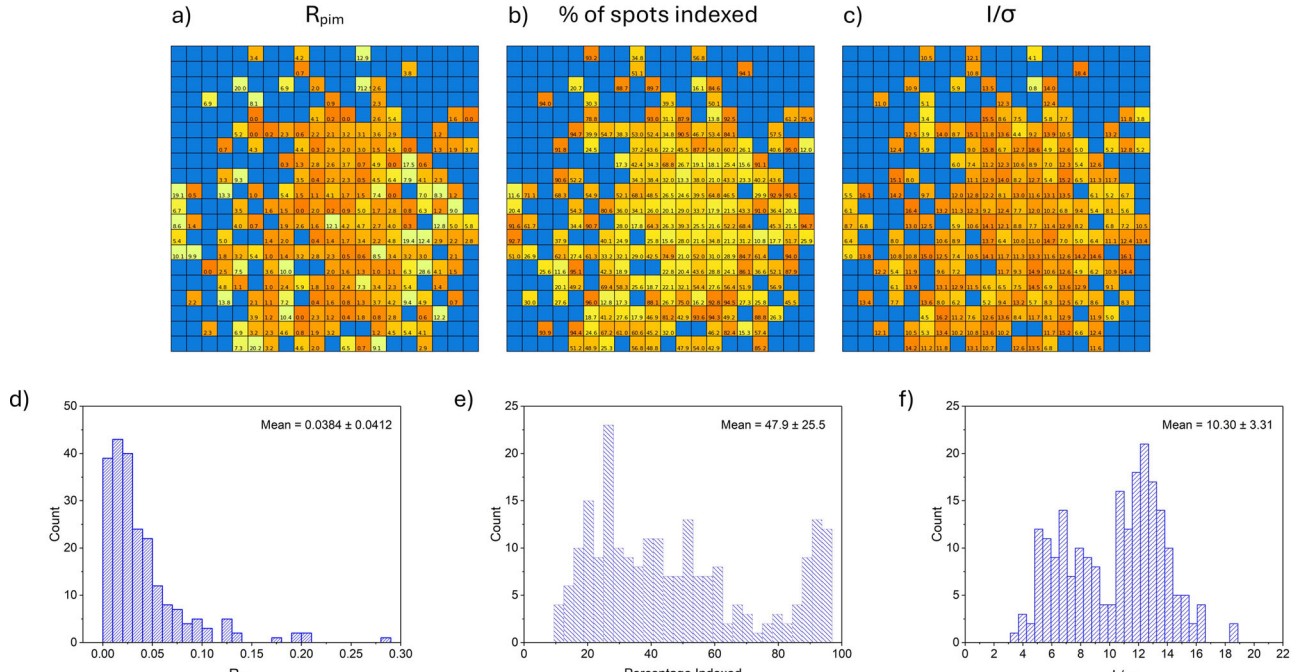

**Fig. 3 | SR-FT-SSX data processing.** Visual representations of the fixed-target grid well positions in real space mapped to their dataset quality factor values, i.e. selection criteria used to select the most successful partial datasets, in the iterative processing route used to generate structure **2**: **a** $R_{pim}$ (yellow = high (poor) value, orange = low (good) value, blue = no value/crystal), **b** % of spots indexed (yellow = low (poor) % value, orange = high (good) % value and blue = no value/crystal) and **c** $I/\sigma(I)$ (yellow = low (poor) value, orange = high (good) value, blue = no value/crystal), alongside histogram representations of the structure quality factors, **d** $R_{pim}$, **e** % of spots indexed and **f** $I/\sigma(I)$.

in the processing input file and will be used by DIALS during indexation. For previously unknown crystal structures, Route B utilizes autoindexing procedures within DIALS where the unit cell parameters are determined ab initio. Further details on each type of indexing procedure are provided in Supplementary Note 4. To illustrate the applicability of SR-FT-SSX for ab initio structure determination, the benchmark data from SNP.2H$_2$O crystals were processed via Route B, leading to the selection of 28 partial datasets for scaling and merging into a final dataset. These data were solved and refined using SHELXT[31] and SHELXL[32], respectively, to yield structure **1** (Table 1). The unit cell parameters were determined to a high degree of accuracy with small estimated standard deviations on all values. 28 partial datasets provide a data completeness of 95.4%, which is slightly lower than the International Union of Crystallography (IUCr) recommendation (98.5%), although the data are more than sufficient for the determination of atomic connectivity.

The data completeness might be increased by relaxing the cut-off values used for the partial dataset selection criteria, but this would reduce the overall quality of the final merged dataset. Instead, we found that an additional processing step can be used to refine the unit cell parameters used for final structure solution, which in turn leads to a significant increase in the data completeness and overall data quality. As shown in Scheme 1, on completion of the first pass through indexing procedures via Route B, the unit cell parameters determined ab initio by autoindexing can be used as the input parameters for a second indexation iteration of the whole grid via Route A. To illustrate the benefit of this iterative approach, the data collected for the benchmark SNP.2H$_2$O crystals were processed in two ways: structure **1** being solved from data processed in a single step via Route B, and structure **2** solved from data processed via the iterative procedure (Route B followed by refinement through Route A).

Both **1** and **2** have been solved only from partial datasets that meet the same selection criteria cut-offs for $R_{pim}$, $I/\sigma(I)$, and the % of spots indexed, and the number of usable partial datasets increased from 28 for **1** to 66 for **2** as a result of the iterative processing step. The visual maps showing the variation in the selection criteria values across the grid for the two processing routes (Fig. S10 and Fig. 3) illustrate how the iterative approach generates considerably more partial datasets that are suitable for merging and this in

turn delivers improved data completeness (99.3%) and redundancy for **2** (Table 1). The increased quality of the final combined crystal structure for **2** compared to **1** is further evidenced by the improvement in other indicators including the signal-to-noise (mean $I/\sigma(I)$) and the residual factors for the final combined structure model ($R_1$, $wR_2$ and $R_{int}$). The final parameters for Structure **2** compare well to those obtained from a single-crystal of the same material on a laboratory diffraction instrument (Table S1), showing that the SR-FT-SSX method can provide high-quality diffraction data to better than atomic-scale resolution, for an inorganic small molecule material.

## Conclusions

In summary, the SR-FT-SSX method has been successfully implemented on Beamline I19 at Diamond Light Source to study the benchmark inorganic material SNP.2H$_2$O. A high-quality X-ray crystal structure with better-than atomic scale resolution, strong signal-to-noise and high completeness was obtained from the merging of 66 datasets and, crucially, our results illustrate how the methodology can be used to determine the structures of unknown small molecule crystal systems. This result is particularly important in light of the continued development of synchrotron and XFEL Light Sources around the globe. The development of methodologies, such as this, to determine small molecule crystal structures by multi-crystal approaches has the potential to unlock SX for organic, inorganic and organometallic samples, allowing access to the structures of ever-smaller microcrystals and circumventing the effects of X-ray damage for a broader range of crystalline materials.

It should be noted that the number of individual crystals required to produce a high quality and complete SR-FT-SSX dataset is considerably fewer than for SFX approaches at XFEL sources, where typically many thousands of crystals are required to offset the effects of fluctuations in the experimental conditions (e.g. crystal size variation and probe beam stability). Here, our approach allows us to mitigate these effects while combining the data from fewer crystals because we collect multiple diffraction images from the same crystal. Thus, we are able to utilize routine single-crystal X-ray diffraction scaling operations on each partial dataset, in addition to the final multi-dataset scaling and merging procedures performed to generate the final dataset. This approach readily accounts for any such variations in

the experiment conditions, producing SSX datasets of comparable quality to a standard single-crystal X-ray collection.

Furthermore, although we collected a total of 50 images from a 5° rotation in this representative experiment, our post-processing tests indicate that a minimum rotation of 10 images (1° rotation, 1 s total X-ray exposure per well) was more than sufficient to obtain good quality unit cell parameters via autoindexing in Route B, while performing the processing steps through DIALS manually provided acceptable unit cell parameters for well-diffracting crystals from as few as just 3 images (0.3° rotation, 0.3 s total X-ray exposure per well). Further details of manual data processing with DIALS with fewer frames from each well is provided in Supplementary Note 6 and Table S2. This offers encouragement that SR-FT-SSX could be used to collect SSX data even from crystals that are very susceptible to radiation damage from the synchrotron X-ray beam, providing structures that are otherwise inaccessible by traditional single-crystal diffraction methodologies. While our approach is unlikely to be directly transferable to diffract-and-destroy serial femtosecond crystallography (SFX) at XFEL facilities, where most crystals are destroyed by a single pulse from the XFEL beam, it is possible that some very robust systems might benefit from a very small rotation and an adaptation of such rotative approaches could be considered to address the ongoing challenge of obtaining SFX data on crystals with small unit cells in the future.

## Methods

### Test crystal system, crystallization and preparation of stock 1-butanol crystal suspensions

Sodium nitroprusside dihydrate ($SNP.2H_2O$) was purchased from Thermo Fisher Scientific and used as received without further purification. The crystal structure of $SNP.2H_2O$ is previously reported with unit cell parameters of $a = 6.17(3)$ Å, $b = 11.84(6)$ Å, $c = 15.43(8)$ Å and space group *Pnnm*, at ambient temperature[29], and its comparatively small unit cell parameters and sparse diffraction pattern (due to the systematic absences created by the *n*-glide symmetry) make it a suitable test-case crystal system for the general development of small molecule SX approaches.

Microcrystal batches of $SNP.2H_2O$ were crystallized using an antisolvent crash crystallization procedure. 100 μL of a saturated solution of $SNP.2H_2O$ in deionized water (0.44 g/g, 0.044 g, 0.15 mmol) held at 40 °C was quickly pipetted into 1000 μL of acetonitrile antisolvent, stirring at 100 RPM. After 5 min to allow for crystallization, the stirring was stopped and the crystals in their mother liquor were pipetted into a 1.5 mL microcentrifuge tube and allowed to settle naturally as a pellet at the base of the tube. To produce 1-butanol suspensions for grid loading, the supernatant was then carefully removed to avoid agitating the crystal pellet, before adding 1 mL of 1-butanol antisolvent. The crystals were then resuspended by sealing and repeatedly inverting the tube before sample delivery onto the grid. Further details of the sample preparation and grid loading procedures are available in Supplementary Note 2.

### Fourier-transform infrared (FT-IR) spectroscopy

FT-IR measurements were performed on a Shimadzu IRAffinity-1S FT-IR Spectrophotometer. Data were collected in % transmittance mode using Happ-Genzel apodization scanning over the range of 500–4000 $cm^{-1}$ and the data averaged over 32 scans.

### Powder X-ray diffraction (PXRD)

Powder X-ray diffraction (PXRD) data were conducted on a Rigaku XtaLab Synergy-R rotating anode diffractometer equipped with a CuKα ($λ = 1.54056$ Å, mirror monochromated) source scanning over a 2θ range of 10° to 40°. Data were recorded at ambient temperature (20 °C), with a small amount of powder (ca. 5 mg) placed onto a Kapton loop with a minimal amount of Fomblin oil.

### Scanning electron microscopy (SEM)

SEM images were collected on a JSM-IT100 InTouchScopeTM Scanning Electron Microscope with a secondary electron detector (SED) set at a working distance of 11 mm, and a probe current of 40 (a.u.) under high vacuum. Crystals observed were prepared onto a silicon nitride fixed-target serial grid following the procedure outlined before being fastened and secured onto a SEM stage.

### Laboratory single crystal X-ray diffraction (SCXRD) data collection

Laboratory SCXRD data for comparative purposes were collected using a Rigaku XtaLab Synergy-R rotating anode diffractometer equipped with a CuKα ($λ = 1.54056$ Å, mirror monochromated) source and a HyPix-3000HE hybrid pixel array detector. The crystal temperature was controlled using an Oxford Instruments Cryojet-XL liquid nitrogen cooling device.

### Small-rotative fixed-target serial synchrotron crystallography (SR-FT-SSX) data collection

Serial crystallography experiments were performed on Beamline I19 at Diamond Light Source using a dual air-bearing fixed-χ diffractometer equipped with a Dectris Pilatus 2 M pixel-array photon-counting detector. The X-ray wavelength was at Ag-edge ($λ = 0.4859$ Å, 25.5140 keV) and a detector distance of 160 mm was used. The 400-well serial grid was mounted on the diffractometer using a modified magnetic base, manually orientated to be approximately normal to the X-ray beam path. The grid position was manipulated using three linear piezo stages from PM-Bearings (RTP-1510-0.1 micron) are mounted on the air-bearing phi ($φ$) axis, which has a 40 °/s rotation speed. The grid was rotated through a small angle at each crystal position by the rotation of the $φ$-axis, as described above, and comprehensive details of the data collection procedure are provided in Supplementary Note 3. The data processing procedures are described in detail both in the main text and in the Supplementary Information (Supplementary Note 4).

## Data availability

Crystallographic data for the structures reported in this article have been deposited at the Cambridge Crystallographic Data Centre, under deposition numbers CCDC 2380952 – 2380954. Copies of the structure data can be obtained free of charge via https://www.ccdc.cam.ac.uk/structures/. Due to their size, raw data frames for all the SR-FT-SSX datasets presented are archived by Diamond Light Source and are freely available on reasonable request. All other relevant data generated and analyzed in this study, including additional methods, spectroscopic data, crystallographic and computational data, are available in this article and its supplementary information in the following formats: 1. Supplementary Notes 1–6[PDF]. 2. Supplementary Movie 1[AVI]. or via the Cardiff University Research Data Repository (Figshare) at https://doi.org/10.17035/cardiff.27372540.

## Code availability

Code for data collection and processing procedures is provided as part of the SR-FT-SSX package and is available at https://github.com/SGLXRD/Hatcher_Group/tree/main. The I19 Serial GUI is under continued development and a beta version of the software can be accessed from the authors on reasonable request.

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

## Acknowledgements
L.E.H. and B.A.C. acknowledge the support of the Royal Society through the funding of LEH's University Research Fellowship (URF\R1\191104). SGL is grateful to Cardiff University, Diamond Light Source and the UK Hub for the Physical Sciences on XFELS (HPSX) for PhD studentship support. The authors thank Diamond Light Source for the award of synchrotron beamtime in experiments (CY33014 and NT35357) and are extremely grateful to several colleagues at Diamond for their advice and input to SX data collection and processing protocols, including Beamline I24 scientists Dr Robin Owen, Dr Danny Axford and Dr Sam Horrell (now Imperial College London); and software scientists Dr Graeme Winter, Dr Nicholas Devenish and Dr Naomi Frisina.

## Author contributions
L.E.H. and M.R.W. contributed equally to this work and are jointly responsible for the conceptualization, methodology, supervision and funding acquisition of the research, as well as writing, reviewing and editing the manuscript. SGL is responsible for methodology, crystallization studies, formal analysis, software development and writing of the original manuscript draft. BAC contributed to software development, formal analysis, visualization of the data and reviewing and editing of the manuscript. A.J.W. is responsible for data collection and reviewing and editing of the manuscript.

## Competing interests
The authors declare no competing interests.
