## [Peer review file · Communications Chemistry]

Small-Rotative Fixed-Target Serial Synchrotron Crystallography (SR-FT-SSX) for Molecular Crystals

Corresponding Author: Dr Lauren Hatcher

Version 0:

Reviewer comments:

Reviewer #1

(Remarks to the Author)

The authors developed a new experimental technique called small-rotative fixed-target serial synchrotron crystallography (SR-FT-SSX), an enhanced method of applying SSX. This approach combines aspects of two pre-existing methods: the crystal rotation technique widely used in single-crystal crystallography and the fixed-target scheme from serial femtosecond crystallography. Although this method is not entirely novel, it offers significant utility for researchers conducting synchrotron-based serial crystallography, particularly those aiming to determine the crystal structures of small organic and inorganic molecules. The authors demonstrated the practical value of this technique by successfully applying it in experiments conducted at a synchrotron. Additionally, they provided several examples illustrating the amount of data required for successful data processing. These efforts will assist readers in applying SR-FT-SSX to their own experiments. Considering this, I recommend the publication of this manuscript in Communications Chemistry. Nevertheless, the authors should consider the following suggestions for further improvement of this manuscript.

1. In the case of serial femtosecond crystallography, variations in conditions that arise during data collection, such as crystal size or X-ray flux, are compensated by averaging the data obtained from more than thousands of crystals. While the experimental setup at a synchrotron is generally more stable, variations in crystal size still play a significant role even in synchrotron-based experiments. Nevertheless, in the example presented by the authors, fewer than 100 crystals were used for data merging, which is significantly less. How many crystals are necessary to offset the effects of fluctuation in the experimental conditions? Are there any guidelines or criteria for this? If guidelines are available, I recommend including the relevant content.
2. As shown in Figure S10, the unit cell parameters obtained through FFT exhibit a fairly broad distribution. For instance, the c-axis shows a distribution ranging from approximately 12 to 17 Å. Were all the results considered reasonable, at least in terms of unit cell parameters, or were only those within a specific range accepted as valid indexing outcomes? Additionally, there are cases where the lengths of the a and b axes were swapped. Was any additional treatment applied during the indexing or merging process in such cases? If so, I recommend including the relevant content.
3. Although the authors do not explicitly state it, they obtained the crystal structure of SNP.2H₂O. I recommend presenting the crystal structure derived from SR-FT-SSX and comparing it with the structure obtained through single-crystal crystallography.

Minor comments

1. The numbers and letters in Figure 3 are hardly visible. It would be helpful to increase the font size, if possible. It might be helpful to include a color bar for Figures 3a–c to improve clarity.
2. Page 6, 280th line: replace “Figure 3a–d” with “Figure 3a–c”.

Reviewer #2

(Remarks to the Author)

I encourage acceptance of this manuscript without change. This paper discusses the challenges and advancements in studying smaller crystals at ultrafast timescales using ultrabright X-ray sources. The increasing brilliance of these sources enables detailed structural analysis but raises the risk of X-ray-induced damage. Serial crystallography (SX) has emerged as a solution, allowing data from multiple crystals to be combined before significant damage occurs. However, SX for small-molecule crystals is difficult due to sparse diffraction patterns, making it hard to determine crystal orientation and combine data. For example, recent results with XFELs and other such light sources have dominated recent literature. These are very detailed and complicated experiments. To address this, the authors present a new methodology—Small-Rotative Fixed-

Target Serial Synchrotron Crystallography (SR-FT-SSX)—which rotates each crystal through a small angle, enabling better data collection and ab initio structure determination. The method is validated on sodium nitroprusside dihydrate microcrystals, achieving atomic-scale resolution ($d_{\min}=0.6 \text{ \AA}$) using only 66 partial datasets. This approach opens new possibilities for high-quality structural analysis of small-molecule crystals in SX.

This work is well-structured and the experiments are well posed. Experimentally, the method involves the collection of crystals on grids. The authors do a good job collecting supporting data of the crystals on the "city block" style grids, which their instrument at the beamline is then able to scan between. Random orientations of the crystals can be collected in each well. To account for the beam sensitivity, small rotations are used, and these are combined.

The authors do an excellent job describing the workflow, and the photographs will be highly useful to those who intend either to pursue this work at the beamline or to replicate related setups at other beamlines. This reviewer was convinced by the data quality pursued. 50 images from a 5° rotation in the representative experiment, postprocessing tests revealed that they really only needed a fraction of the data that was collected.

I would have liked to see true unknowns obtained but this is more than sufficient to prove and demonstrate the method and I do not believe the manuscript would need to demonstrate such a thing to warrant publication. I would like to see the manuscript in print to begin citing immediately; it will be cited in 4 of my upcoming works.

Reviewer #1

The authors developed a new experimental technique called small-rotative fixed-target serial synchrotron crystallography (SR-FT-SSX), an enhanced method of applying SSX. This approach combines aspects of two pre-existing methods: the crystal rotation technique widely used in single-crystal crystallography and the fixed-target scheme from serial femtosecond crystallography. Although this method is not entirely novel, it offers significant utility for researchers conducting synchrotron-based serial crystallography, particularly those aiming to determine the crystal structures of small organic and inorganic molecules. The authors demonstrated the practical value of this technique by successfully applying it in experiments conducted at a synchrotron. Additionally, they provided several examples illustrating the amount of data required for successful data processing. These efforts will assist readers in applying SR-FT-SSX to their own experiments. Considering this, I recommend the publication of this manuscript in Communications Chemistry. Nevertheless, the authors should consider the following suggestions for further improvement of this manuscript.

We thank the reviewer for their careful consideration of our manuscript and their positive recommendation.

1. In the case of serial femtosecond crystallography, variations in conditions that arise during data collection, such as crystal size or X-ray flux, are compensated by averaging the data obtained from more than thousands of crystals. While the experimental setup at a synchrotron is generally more stable, variations in crystal size still play a significant role even in synchrotron-based experiments. Nevertheless, in the example presented by the authors, fewer than 100 crystals were used for data merging, which is significantly less. How many crystals are necessary to offset the effects of fluctuation in the experimental conditions? Are there any guidelines or criteria for this? If guidelines are available, I recommend including the relevant content.

The reviewer is indeed correct that, in the corresponding serial femtosecond experiment at XFEL facilities it is necessary to collect data on 1000s of crystals to be able to offset the effects of, often significant, fluctuations in the experimental conditions. This is because, in these SFX experiments where only a single frame of data can be collected from any one crystal, it is very difficult to find a consistent route to scale the images from different exposures made on different crystals.

Our method can mitigate any fluctuations that do occur as we collect *multiple* images from the same crystal in our small rotation partial datasets. As we treat each of these partial datasets essentially as if they are a standard single-crystal X-ray diffraction dataset, we are able to perform routine scaling operations through the DIALS suite (using `dials.scale`) on *each* of these partial datasets, even before the best datasets are selected for merging and re-scaling into the final combined dataset. This dual-scaling approach (both on the individual crystal, and then again at the stage of merging the data from multiple crystals) allows us to readily account for fluctuations in the diffraction data caused by experimental considerations such as small variations in the synchrotron beam intensity or crystal size.

We agree with the reviewer that this point is not well described in either the Manuscript or in Supplementary Note 4, and so we have made the following amendments to the text in each document:

Revised Manuscript:

- [Text amended on page 6, lines 13-15] *Once suitable unit cell parameters and a space group are determined, integration **and scaling** procedures are then initiated for all viable partial datasets within DIALS, providing intensity data for individual wells. Once **partial dataset processing procedures** are complete...*
- [Text amended on page 6, lines 24-25] *Once identified, the best partial datasets are **re-submitted to DIALS to facilitate multi-dataset scaling and merging procedures through `DIALS.scale`**,³⁰ to produce a final set of combined diffraction intensity data in the required format for structure solution and refinement.*
- [Paragraph added to the Conclusions, Page 8, lines 31-38] ***It should be noted that the number of individual crystals required to produce a high quality and complete SR-FT-SSX dataset is considerably fewer than for SFX approaches at XFEL sources, where typically many thousands of crystals are required to offset the effects of fluctuations in the experimental conditions (e.g. crystal size variation and probe beam stability). Here, our approach allows us to mitigate these effects while combining the data from fewer crystals because we collect multiple diffraction images from the same crystal. Thus, we are able to utilize routine single-crystal X-ray diffraction scaling operations on each partial dataset, in addition to the final multi-dataset scaling and merging procedures performed to generate the final dataset. This approach readily***

accounts for any such variations in the experiment conditions, producing SSX datasets of comparable quality to a standard single-crystal X-ray collection.

Supplementary Note 4:

- [Text added to Page 8, lines 28-31] *By running `Chipreader.py`, the user submits each partial dataset collected on the individual crystal through a routine DIALS processing regime, which is treated as if it is a standard (though incomplete) single crystal X-ray data collection. Thus, each partial dataset is first subject to spot (reflection) finding, indexing (through the specified mode of indexing as described above), integration and scaling procedures.*
- [Text amended on Page 10, lines 10-15] *The merged dataset is produced by calling the `DIALS.scale` command towards the selected partial datasets from the chosen wells, which configures a “Multiscaler” regime to scale the individual datasets being merged to account for any variation in the diffraction data due to experimental factors such as crystal-to-crystal size variations or fluctuations in the synchrotron beam. Scaling of multiple datasets from different crystals is possible for our method because within our partial datasets collection on individual crystals we have sufficient coverage of reciprocal space, and good sampling of the reflection profiles due to our thin-slicing rotative approach. An example output from the Multiscaler merging process is archived with the data processing scripts online and can be accessed via the Figshare link provided in the Data Availability statement. Once complete, standard SHELX `.INS` and `.HKL` files are output that can be reprocessed through a space group determination program of choice.*

2. As shown in Figure S10, the unit cell parameters obtained through FFT exhibit a fairly broad distribution. For instance, the c-axis shows a distribution ranging from approximately 12 to 17 Å. Were all the results considered reasonable, at least in terms of unit cell parameters, or were only those within a specific range accepted as valid indexing outcomes? Additionally, there are cases where the lengths of the a and b axes were swapped. Was any additional treatment applied during the indexing or merging process in such cases? If so, I recommend including the relevant content.

The reviewer is correct that there is a considerable variation in the quality of the indexed information obtained from different wells through the FFT indexation mode (where no prior information is provided by the user). As discussed in the Manuscript and in Supplementary Note 4, the best partial datasets are chosen according to consideration of a number of structure quality descriptors, and for SNP.2H₂O the most informative values were found to be R_{pim} , % spots indexed and $I/\sigma(I)$ value. For a true unknown, once the data from the chip are assessed and filtered according to the most suitable quality factors by using the `chipreader.py` script, then the user is required to assess the most common unit cell parameters that occur in these partial datasets. It is possible to amend the `chipreader.py` script to output visual plots of the spread of unit cell parameters indexed, though this is not presented here as it was not necessary for the current investigation.

We emphasize in the manuscript that we recommend the next step to be that the serial grid is then re-processed through the real-space-grid-search indexing method, applying those most common unit cell parameters as known values in this second processing iteration.

We have added an amendment to page 7, line 18 of the Revised Manuscript that reinforces that this second iterative step should still be applied to the *whole* grid:

“As shown in Scheme 1, on completion of the first pass through indexing procedures via Route B, the unit cell parameters determined ab initio by autoindexing can be used as the input parameters for a second indexation iteration of the whole grid via Route A.”

This is the only additional treatment that was applied during indexing and merging. Our iterative approach allows the user to further refine the data, leading to an increase in the number of suitable partial datasets and greater overall completeness and final structure quality.

We believe that the implementation of this iterative process is the key to ensuring the correct unit cell parameters are identified and refined to optimal values. It may be the case that initially the FFT approach will identify sub-optimal parameters or, as the reviewer suggests, swap around axes for a high-symmetry space group. I would note that even during a standard single crystal data collection when indexing is performed on-the-fly in commercial software then often similar discrepancies are seen, thus no individual indexing route is entirely foolproof and the output is always improved by identifying and including more, good quality data. We are confident that the 2nd iteration of data processing really helps to refine the cell parameters and further filters out any bad datasets, which tends to eliminate most of the (incorrect) satellite values of cell parameters.

In the hypothetical case that this methodology was used to investigate a sample with a bimodal distribution of unit cells, e.g. a 2-phase physical mixture such as a polymorphic system may produce, as the method currently stands then the user would need to spend some time running the FFT process, identifying the most commonly observed unit cell parameters, and then iterating through the real-space step until the correct unit cell parameters for each phase were confirmed. The investigation of such a system is currently underway in our group and new routes to streamline this process are currently under development. Thus, a report of the best practice in dealing with such a mixture of different, and possibly unknown, crystal forms will be forthcoming in future publication.

3. Although the authors do not explicitly state it, they obtained the crystal structure of SNP.2H₂O. I recommend presenting the crystal structure derived from SR-FT-SSX and comparing it with the structure obtained through single-crystal crystallography.

This comparison is already made in the main Manuscript on Page 7, line 30. A comparative single-crystal X-ray data collection was run on a representative plate-like SNP.2H₂O crystal in-house using a high-flux Rigaku Synergy-R rotating anode instrument, and the summary table of the data (for comparison to the SR-FT-SSX data) is provided in Table S1 in Supplementary Note 4.

Minor comments

1. The numbers and letters in Figure 3 are hardly visible. It would be helpful to increase the font size, if possible. It might be helpful to include a color bar for Figures 3a–c to improve clarity.

It is difficult to increase the size of the text in Figures 3a–c due to the size of the grid positions. Instead we have opted to revise the figure caption for both this figure and Figure S9. We are happy to take the advice of the editor on the required size of the individual images in Figure 3 and will provide the high-quality files of each image a – f.

The Figure caption for Figure 3 in the Revised Manuscript now reads:

Figure 3. Visual representations of the fixed-target grid well positions in real space mapped to their dataset quality factor values i.e. selection criteria used to select the most successful partial datasets) in the iterative processing route used to generate structure 2: a) R_{pim} (yellow = high (poor) value, orange = low (good) value, blue = no value/crystal), b) % of spots indexed (yellow = low (poor) % value, orange = high (good) % value and blue = no value/crystal) and c) $I/\sigma(I)$ (yellow = low (poor) value, orange = high (good) value, blue = no value/crystal), alongside histogram representations of the structure quality factors d) R_{pim} , e) % of spots indexed and f) $I/\sigma(I)$.”

2. Page 6, 280th line: replace “Figure 3a–d” with “Figure 3a–c”.

We thank the reviewer for the careful reading of our manuscript and have made the required change in the Revised Manuscript.

Reviewer #2

I encourage acceptance of this manuscript without change. This paper discusses the challenges and advancements in studying smaller crystals at ultrafast timescales using ultrabright X-ray sources. The increasing brilliance of these sources enables detailed structural analysis but raises the risk of X-ray-induced damage. Serial crystallography (SX) has emerged as a solution, allowing data from multiple crystals to be combined before significant damage occurs. However, SX for small-molecule crystals is difficult due to sparse diffraction patterns, making it hard to determine crystal orientation and combine data. For example, recent results with XFELs and other such light sources have dominated recent literature. These are very detailed and complicated experiments. To address this, the authors present a new methodology—Small-Rotative Fixed-Target Serial Synchrotron Crystallography (SR-FT-SSX)—which rotates each crystal through a small angle, enabling better data collection and ab initio structure determination. The method is validated on sodium nitroprusside dihydrate microcrystals, achieving atomic-scale resolution ($d_{\text{min}}=0.6 \text{ \AA}$) using only 66 partial datasets. This approach opens new possibilities for high-quality structural analysis of small-molecule crystals in SX.

This work is well-structured and the experiments are well posed. Experimentally, the method involves the collection of crystals on grids. The authors do a good job collecting supporting data of the crystals on the “city block” style grids, which their instrument at the beamline is then able to scan between. Random orientations of the crystals can be collected in each well. To account for the beam sensitivity, small rotations are used, and these are combined.

The authors do an excellent job describing the workflow, and the photographs will be highly useful to those who intend either to pursue this work at the beamline or to replicate related setups at other beamlines. This reviewer was convinced by the data quality pursued. 50 images from a 5° rotation in the representative experiment, postprocessing tests revealed that they really only needed a fraction of the data that was collected.

I would have liked to see true unknowns obtained but this is more than sufficient to prove and demonstrate the method and I do not believe the manuscript would need to demonstrate such a thing

to warrant publication. I would like to see the manuscript in print to begin citing immediately; it will be cited in 4 of my upcoming works.

We thank the reviewer for their careful reading of our manuscript and their positive recommendation.

We hope that, with these changes, our manuscript will now be suitable for publication. However, please do not hesitate to contact us should you require any further information. We look forward to hearing from you again in due course.

Yours sincerely,

Dr Lauren E. Hatcher, on behalf of all authors